# Characterization of two non-competing antibodies to influenza H3N2 hemagglutinin stem reveals its evolving antigenicity

Akshita B. Gopal[1,10], Huibin Lv [1,2,10] ✉, Tossapol Pholcharee[1], Wenhao O. Ouyang[1], Qi Wen Teo [1,2], Yasha Luo[3], Yun Sang Tang[4,5], Mingyong Luo[3], Chris K. P. Mok [4,5,6,7] & Nicholas C. Wu [1,2,8,9] ✉

The conserved stem domain of influenza hemagglutinin (HA), which is classified into group 1 and group 2, is a target of broadly neutralizing antibodies. While many group 1 HA stem antibodies have been described, much less is known about group 2 HA stem antibodies. This study structurally characterizes two group 2 HA stem antibodies, 2F02 and AG2-G02, targeting the central stem epitope and the lower stem epitope, respectively. Unlike prototypic antibodies to group 2 HA stem, 2F02 and AG2-G02 do not compete for binding. Both antibodies offer protection in a female mouse model via neutralization activity and Fc-mediated effector functions. We further demonstrate that the natural evolution of HA2 position 32 restricts the binding of AG2-G02 to recent human H3N2 HAs and influences the binding of human plasma samples. Overall, these findings advance our understanding of the antigenicity of HA stem, which has important implications for the development of broadly protective influenza vaccines.

With seasonal influenza A viruses causing around 3–5 million cases of severe illness each year globally, they remain a persistent threat to public health[1]. Hemagglutinin (HA) and neuraminidase (NA) are the major surface glycoprotein antigens of the influenza A viruses. HA and NA are classified into 19 and 11 antigenic subtypes, respectively[1,2]. The 19 HA subtypes are further categorized into two groups. Group 1 HA includes H1, H2, H5, H6, H8, H9, H11, H12, H13, H16, H17, H18 and H19, whereas group 2 HA includes H3, H4, H7, H10, H14, and H15. Among these, H1N1 and H3N2 are responsible for seasonal influenza epidemics in humans. H3N2 was introduced into the human population in 1968 after a reassortment event involving an avian H3 virus[3]. New antigenic variants of human H3N2 viruses emerge every 3–5 years, often resulting in antigenic mismatches between seasonal influenza vaccine

strains and circulating strains[4,5]. Furthermore, spillover events of avian H3 viruses into humans have been reported[6–9]. In China, avian H3N8 viruses were linked to two confirmed human infections in 2022, followed by the first case of mortality in 2023[6–8]. Although seasonal influenza vaccines confer protection against the human H3N2 viruses, the human population lacks immunity to the zoonotic H3 strains[10].

HA is expressed as a single polypeptide chain (HA0), which forms a homotrimer. After protease cleavage, HA0 matures into HA1 and HA2 subunits. The immunodominant head domain, composed entirely of the HA1 subunit, sits atop the membrane-proximal stem domain, primarily composed of the HA2 subunit[11]. The HA head binds to sialylated host receptors and is highly variable, whereas the evolution of the HA stem is somewhat constrained, as it possesses the membrane

[1]Department of Biochemistry, University of Illinois Urbana-Champaign, Urbana, IL, USA. [2]Carl R. Woese Institute for Genomic Biology, University of Illinois Urbana-Champaign, Urbana, IL, USA. [3]Department of Clinical Laboratory, Guangdong Women and Children Hospital, Guangzhou, China. [4]The Jockey Club School of Public Health and Primary Care, The Chinese University of Hong Kong, Hong Kong, China. [5]Li Ka Shing Institute of Health Sciences, Faculty of Medicine, The Chinese University of Hong Kong, Hong Kong, China. [6]S.H. Ho Research Centre for Infectious Diseases, The Chinese University of Hong Kong, Hong Kong, China. [7]School of Biomedical Sciences, The Chinese University of Hong Kong, Hong Kong, China. [8]Center for Biophysics and Quantitative Biology, University of Illinois Urbana-Champaign, Urbana, IL, USA. [9]Carle Illinois College of Medicine, University of Illinois Urbana-Champaign, Urbana, IL, USA. [10]These authors contributed equally: Akshita B. Gopal, Huibin Lv. ✉e-mail: huibinlv@illinois.edu; nicwu@illinois.edu

fusion machinery essential for viral entry. Since the HA stem is highly conserved compared to the HA head[12,13], antibodies targeting the HA stem are often broadly neutralizing against multiple antigenically distinct strains or even subtypes[11]. Studies have also demonstrated that HA stem antibodies have fewer potential escape mutants as compared to the HA head antibodies[14,15]. Therefore, the discovery and characterization of HA stem antibodies have motivated the development of broadly protective influenza vaccines[16–19], with some of these candidates showing promising results in phase I clinical trials[20–22].

While numerous stem-binding antibodies targeting group 1 HA or cross-react with both group 1 and 2 HAs have been identified, those specific to group 2 HA are much less explored[23–25]. Antibodies to group 2 HA stem are mapped to two slightly overlapping yet distinct epitopes[11,26]. The first one is known as the central stem epitope, which is centered at helix A of the HA2 subunit[27]. The second one is known as the lower stem epitope, which is closer to the viral membrane compared to the central stem epitope and involves the basal β-sheet that is formed by both HA1 and HA2 subunits[28,29]. Many central stem antibodies acquire reactivity to group 2 HA from a group 1-specific ancestor[24,30,31]. As a result, most central stem antibodies targeting group 2 HA typically also cross-react with group 1 HA[24,28,30–32], although exceptions exist[25]. By contrast, lower stem antibodies are specific to group 2 influenza virus HAs[26,33]. Due to their larger breadth, central stem antibodies have received a lot more attention than lower stem antibodies. As a result, the contribution of lower stem antibodies to human antibody responses against the influenza virus remains largely elusive.

Recently, we discovered dozens of HA stem antibodies from the literature using machine learning and high-throughput experimental screening[33,34]. Two of these HA stem antibodies, namely AG2-G02[35] and 240-14 IgA 2F02[36] (hereinafter abbreviated as 2F02), are group 2-specific. In this study, cryogenic electron microscopy (cryo-EM) analysis showed that AG2-G02 and 2F02 bound to the lower stem and central stem epitopes, respectively. We further demonstrated that AG2-G02 and 2F02 could bind concurrently to the HA stem. While 2F02 bound to all tested H3 HAs, AG2-G02 did not bind to HAs from the recent human H3N2 strains. We found that a natural mutation at HA2 position 32 of the recent human H3N2 strains abolished the binding of AG2-G02. Additionally, serological analysis indicated that the natural evolution of HA2 position 32 has altered the antigenicity of human H3N2 HA stem. Throughout this study, H3 numbering and Kabat numbering are used for HA residue positions and antibody residue positions, respectively, unless otherwise stated.

## Results

### Identification of two non-competing antibodies to H3 HA stem
We have previously curated a large dataset of human monoclonal HA antibodies, among which 4469 did not have any epitope information[33]. From these 4469 antibodies, we have further identified AG2-G02[35] and 2F02[36], both of which were derived from the plasmablasts of vaccinees, as HA stem antibodies that target H3 HA but not H1 HA[33,34]. AG2-G02 is encoded by *IGHV1-2/IGHD3-3/IGHJ5* and *IGKV3-20/IGKJ3*[33,35], whereas 2F02 is encoded by *IGHV3-23/IGHD3-3/IGHJ5* and *IGLV2-11/IGLJ3*[33,36]

To probe the epitopes of AG2-G02 and 2F02, we performed a binding competition assay using an H3 stem construct designed based on H3N2 A/Finland/486/2004 HA[18]. Our competition assay also included two central stem antibodies, CR9114 and FI6v3, as well as two lower stem antibodies, CR8020 and CR8043[24,28,29,37]. AG2-G02 competed strongly with all the tested antibodies except 2F02 and FI6v3. Similarly, 2F02 competed strongly with all the tested antibodies except AG2-G02 (Fig. 1A). This observation suggested that the epitopes of AG2-G02 and 2F02 were proximal to each other but did not overlap.

### Structural analysis reveals distinct epitopes for AG2-G02 and 2F02
To characterize the epitopes of AG2-G02 and 2F02 and their molecular basis of binding, we determined the cryo-EM structures of AG2-G02 Fab and 2F02 Fab in complex with H3N8 A/mallard/Alberta/362/2017 HA to resolutions of 2.60 Å and 2.71 Å, respectively (Table S1). Structural analysis revealed that AG2-G02 bound to the lower stem epitope, whereas 2F02 bound to the central stem epitope (Fig. 1B). Although the epitope of AG2-G02 highly overlapped with that of CR8020[28] and CR8043[29], it is shifted laterally with its complementarity-determining region (CDR) H3 stretching into a groove along the basal β-sheet (Figs. 1C, S1A). Notably, the epitope of AG2-G02 resembled that of another lower stem antibody, ADI-85666 (Fig. S1C, D), as reported in a recent study[38]. Although 2F02, CR9114, and FI6v3 target a similar epitope (Fig. 1C), their differences in the angles of approach explain why CR9114, but not 2F02 and FI6v3, competed with AG2-G02 (Fig. 1D, E). Consistently, low-resolution cryo-EM analysis showed that three copies of AG2-G02 Fab and three copies of 2F02 Fab could concurrently bind to a single HA trimer (Fig. 1F).

### Both AG2-G02 and 2F02 use *IGHD3-3*-encoded CDR H3 for binding
Both AG2-G02 and 2F02 relied heavily on CDR H3 for binding to HA, with greater than 40% of the paratope buried surface area attributing to CDR H3 (Fig. 2A, B). Besides, their CDR H3s shared an *IGHD3-3*-encoded DFW motif, albeit engaging HA very differently (Figs. 2C, S2). For the DFW motif (positions 98–100) in 2F02, the side chain of $V_H$ W100 H-bonded with the backbone oxygen of $T318_{HA1}$ and formed a T-shaped π-π stacking interaction with $W21_{HA2}$. By contrast, for the DFW motif (positions 97–99) in AG2-G02, the side chain of $V_H$ W99 did not interact with HA. Instead, AG2-G02 relied on the side chain of $V_H$ F98 in the DFW motif for binding via a cation-π stacking interaction with $R25_{HA2}$. Notably, neither AG2-G02 nor 2F02 used the side chain of the Asp in the DFW motif for binding. AG2-G02 and 2F02 are also H-bonded with HA through the side chains of multiple other residues. AG2-G02 H-bonded with HA via the side chains of multiple tyrosines on the heavy chain, including $V_H$ Y32, $V_H$ Y96, and $V_H$ Y100c, whereas 2F02 H-bonded with HA via the side chains of $V_H$ S52, $V_H$ N100f, $V_L$ K52, and $V_L$ K53. These observations indicated that although AG2-G02 and 2F02 converged on the use of the *IGHD3-3*-encoded DFW motif, they targeted distinct epitopes with different binding modes.

### AG2-G02 and 2F02 are neutralizing antibodies that broadly react with H3 strains
To examine the cross-reactive breadth of AG2-G02 and 2F02, we used an enzyme-linked immunosorbent assay (ELISA) to measure their binding activity against various group 2 HAs. We found that 2F02 bound to all tested H3 HAs and cross-reacted with an H15 HA (Fig. 3A). By contrast, the reactivity breadth of AG2-G02 was restricted to H3 HAs. Additionally, AG2-G02 had no detectable binding to HAs from more recent human H3N2 strains, including H3N2 A/Switzerland/9715293/2013 and H3N2 A/Darwin/6/2021. Consistently, while AG2-G02 and 2F02 had similar binding activity to cells infected with H3N2 A/Moscow/10/1999, the binding activity of AG2-G02 to cells infected with the H3N2 A/Darwin/6/2021 virus was much weaker than that of 2F02 (Fig. S3A, B). Although the reactivity breadth of 2F02 was broader than AG2-G02, none of them reacted with H7 HAs, unlike CR8020[28] and CR8043[29].

We further assessed the neutralization activity of AG2-G02 and 2F02. While 2F02 demonstrated measurable neutralization activity against all five tested H3 strains spanning from 1982 to 2021 (Fig. 3B), AG2-G02 only neutralized four and not H3N2 A/Darwin/6/2021, consistent with its lack of binding to this strain (Fig. 3A, B). Notably, the neutralization activity of AG2-G02, 2F02, and our positive control antibodies CR8020 and CR8043 was much weaker when they were not

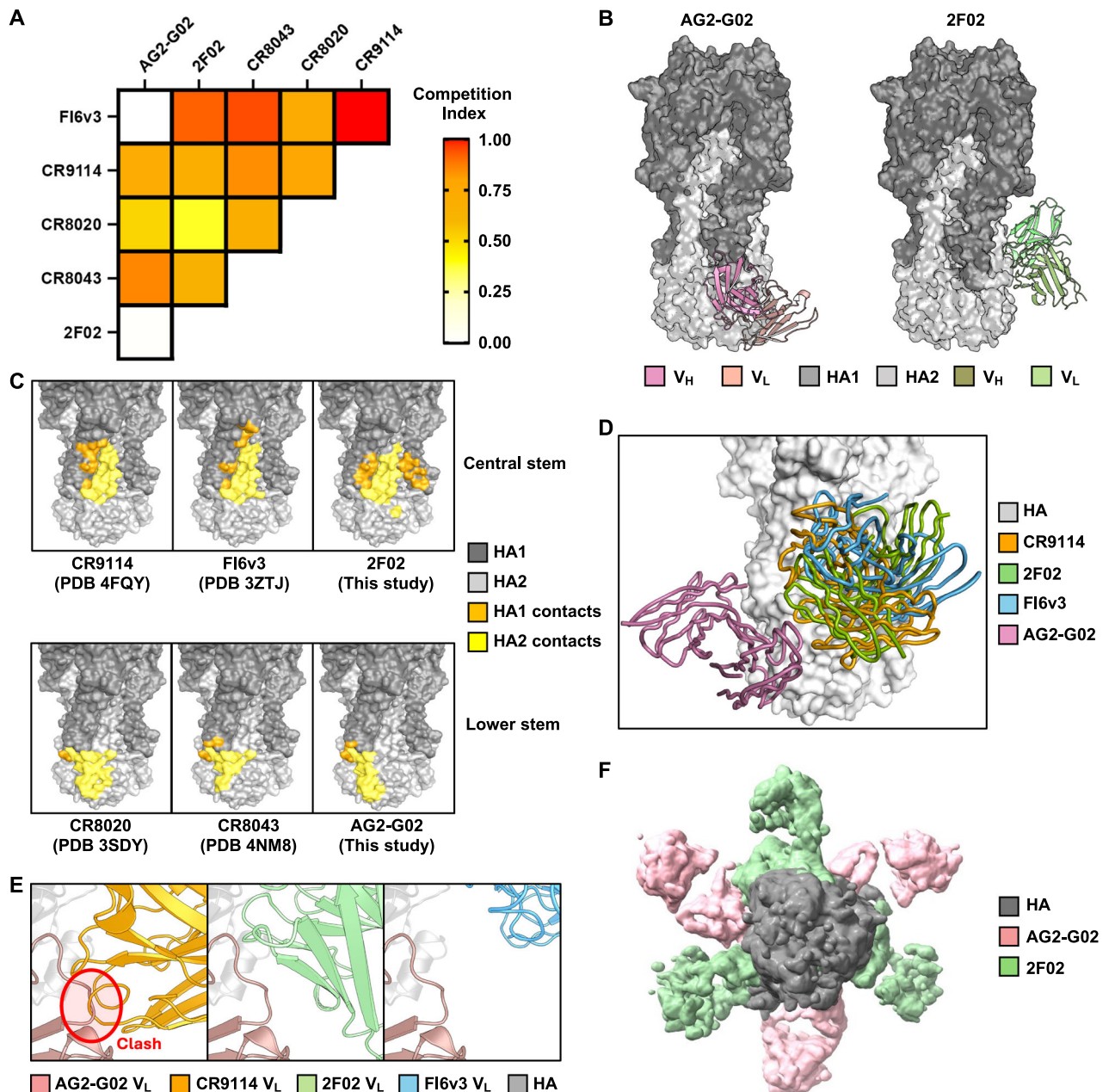

**Fig. 1 | AG2-G02 and 2F02 have non-overlapping epitopes. A** Competition among the indicated antibodies for binding to the H3 stem was measured by biolayer interferometry (BLI). All antibodies were in Fab format except for FI6v3, which was in IgG format. Competition index is shown as a heatmap. **B** Cryo-EM structures of AG2-G02 and 2F02 in complex with H3N8 A/mallard/Alberta/362/2017 HA are shown. HA1 and HA2 are in dark gray and light grey, respectively. The variable heavy ($V_H$) and light chains ($V_L$) of AG2-G02 are in dark pink and pale pink, respectively. The variable heavy ($V_H$) and light chains ($V_L$) of 2F02 are in dark green and pale green, respectively. **C** Epitopes of CR9114 (PDB 4FQY)[37], FI6v3 (PDB 3ZTJ)[32], 2F02 (this study), CR8020 (PDB 3SDY)[28], CR8043 (PDB 4NM8)[29], and AG2-G02 (this study) are compared. Epitope residues on HA1 and HA2 are in orange and yellow, respectively. **D** The binding of AG2-G02 (pink) and CR9114 (orange), 2F02 (green), and FI6v3 (blue) to HA (grey) is compared. **E** A close-up view illustrating a steric clash between variable light chain ($V_L$) regions of AG2-G02 (pink) and CR9114 (orange). Such a steric clash was not observed between AG2-G02 and 2F02 (green) or between AG2-G02 and FI6v3 (blue). **F** A top-down view of AG2-G02 (pink) and 2F02 (green) Fabs simultaneously in complex with an HA trimer (grey), which was determined to a resolution of 5.6 Å using cryo-EM.

maintained in the medium throughout the assay (Fig. 3C), consistent with the known ability of HA antibodies in inhibiting viral release[39–43]. Together, these results demonstrated that both AG2-G02 and 2F02 are neutralizing antibodies with broad reactivity against H3 HAs.

## AG2-G02 and 2F02 confer in vivo protection against influenza virus lethal challenge

To test the in vivo protection activity of AG2-G02 and 2F02, they were administered prophylactically at 5 mg/kg to six-week-old female BALB/c mice, followed by a lethal challenge with H3N2 A/Philippines/2/1982

(X-79) virus. Both AG2-G02 and 2F02 conferred prophylactic protection, as indicated by survival analyses (Fig. 4A), weight loss profiles (Fig. 4B), and lung viral titers on day 3 post-infection (Fig. 4C). To investigate the role of Fc-mediated effector functions in the in vivo protection activity of AG2-G02 and 2F02, a LALA-PG variant was generated for each antibody to eliminate their Fc-mediated effector functions[44]. Mice treated with the LALA-PG variants of both AG2-G02 and 2F02 exhibited reduced survival (Fig. 4A), increased weight loss (Fig. 4B), and higher lung viral titers on day 3 post-infection (Fig. 4C) than those treated with their wild-type counterparts. Additionally,

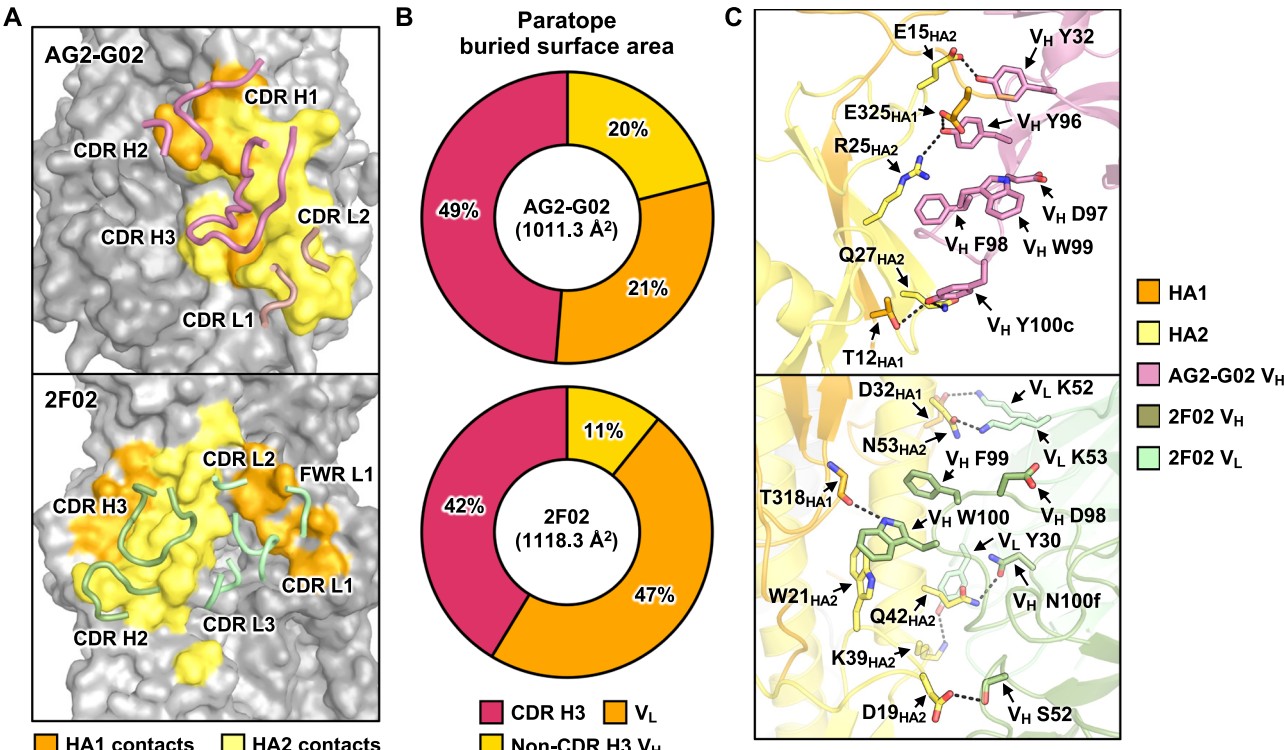

**Fig. 2 | Structural analysis of the binding of AG2-G02 and 2F02 to HA.**
**A** Interaction of CDRs (ribbon representation) of AG2-G02 and 2F02 with HA is shown. **B** Contributions from CDR H3 (red), non-CDR H3 $V_H$ (yellow), and $V_L$ (orange) to the paratope buried surface area (BSA). **C** A close-up view of the molecular interactions between the antibodies and HA. Key interacting residues are shown by sticks representation. Black dashed lines represent H-bonds and salt bridges. **A**, **C** The heavy and light chains of AG2-G02 are in pink and pale pink, respectively. The heavy and light chains of the 2F02 heavy chain are in green and pale green, respectively. Epitope residues on HA1 and HA2 are in orange and yellow, respectively.

replacing the human Fc region with murine Fc demonstrated slightly improved protection, as shown by the negligible weight loss (Fig. S4A, B). Collectively, our results highlight the importance of Fc-mediated effector functions for the in vivo protection activity of AG2-G02 and 2F02. Consistently, both AG2-G02 and 2F02 could elicit antibody-dependent cell-mediated cytotoxicity (ADCC) activity in vitro (Fig. S5). Nevertheless, the neutralization of AG2-G02 and 2F02 also contributes to their in vivo protection activity, given that their LALA-PG variants were still able to confer protection (Fig. 4A–C).

### The antigenicity of the lower stem epitope has evolved in human H3N2 viruses

To investigate the molecular basis for the lack of binding of AG2-G02 to the recent human H3N2 HAs, we examined the natural amino acid variants in the AG2-G02 epitope (Fig. 5A). We found that the amino acid variants at HA2 position 32 of H3 HAs correlated with AG2-G02 reactivity. H3 HAs that reacted with AG2-G02 either possessed a Thr or an Ile at HA2 position 32, whereas those that did not bind AG2-G02 had an Arg (Figs. 5A, S6). When the H3N2 virus entered the human population in 1968, it carried $T32_{HA2}$. During the 2004–2005 influenza season, $T32_{HA2}$ was replaced by $I32_{HA2}$, which was subsequently replaced by $R32_{HA2}$ in the 2008-2009 influenza season (Fig. 5A). Our cryo-EM structure of AG2-G02 in complex with H3 HA showed that $T32_{HA2}$ fitted tightly into a pocket in the AG2-G02 paratope. Structural modeling indicated that $I32_{HA2}$, albeit slightly bulkier than $T32_{HA2}$, would still fit into the pocket (Fig. 5B). However, $R32_{HA2}$ would clash with AG2-G02. Consistently, biolayer interferometry analysis showed that AG2-G02 bound to the H3 stem with $T32_{HA2}$ ($K_d = 0.57$ nM) and $I32_{HA2}$ ($K_d = 0.21$ nM) with similar affinity but did not bind to the H3 stem with $R32_{HA2}$ (Figs. 5B, S7A). As a control, we demonstrated that the binding activity of 2F02 was minimally affected by these H3 stem

variants (Fig. S7B). These results suggested that mutations at HA2 position 32 could influence the antigenicity of the lower stem epitope.

To further analyze how the natural evolution of HA2 position 32 influenced the antigenicity of the HA stem, we measured the binding activity of human plasma samples to different H3 stem variants ($T32_{HA2}$, $I32_{HA2}$, and $R32_{HA2}$). Three groups of plasma samples were obtained. The first group was collected prior to 2003 from 20 adults aged between 20 and 52 (pre-2003 adults) who should have been infected by H3N2 strains with $T32_{HA2}$ but not $I32_{HA2}$ or $R32_{HA2}$ (Fig. S8). The second group was collected after 2011 from 20 adults over the age of 55 (post-2011 adults), whose initial exposure to the human H3N2 virus likely involved strains with $T32_{HA2}$. The third group included 15 infants born after 2011 (post-2011 infants), who, if previously been infected with the human H3N2 virus, would have encountered strains with $R32_{HA2}$ but not $T32_{HA2}$ or $I32_{HA2}$. The binding activity of plasma samples from pre-2003 adults to H3 stem with $T32_{HA2}$ (median AUC = 11,044) was slightly, yet significantly, higher than that with $I32_{HA2}$ (median AUC = 7007, p-value = 0.0001) or $R32_{HA2}$ (median AUC = 9321, p-value = 0.03, Fig. 5C). Similarly, the binding activity of plasma samples from post-2011 adults to H3 stem with $T32_{HA2}$ (median AUC = 25,916) was also slightly, yet significantly, higher than that with $I32_{HA2}$ (median AUC = 17,988, p-value = 0.001, Fig. 5D). By contrast, the binding activity of plasma samples from post-2011 infants to H3 stem with $R32_{HA2}$ (median AUC = 21,083) was significantly higher than that with $T32_{HA2}$ (median AUC = 13,123, p-value = 0.008) or $I32_{HA2}$ (median AUC = 12,415, p-value = 0.0004, Fig. 5E). Notably, the central stem epitope was the same among all three H3 stem variants in this experiment. Therefore, the impact of mutations at HA2 position 32 on the antigenicity of the lower stem epitope should be larger than what was measured in this experiment, due to the contribution of central stem antibodies to the binding signal. Overall, these observations

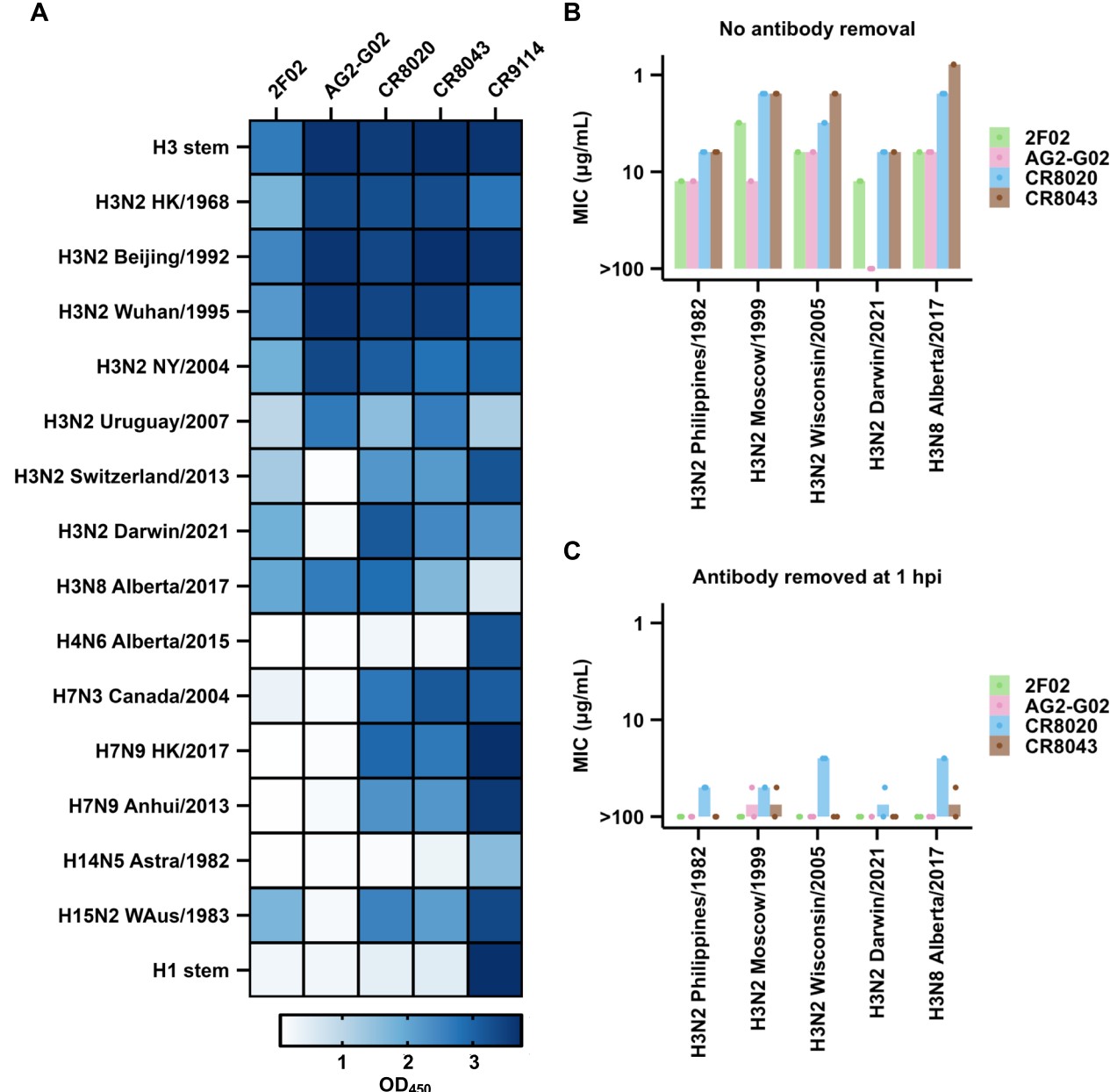

**Fig. 3 | Binding activity and neutralization activity of AG2-G02 and 2F02. A** The binding activities of AG2-G02 and 2F02 against recombinant HA proteins from the specified influenza strains were measured by ELISA. The absorbance values at 450 nm are shown as a heatmap. **B**, **C** The neutralization activity of AG2-G02 and 2F02 against recombinant H3N8 and different recombinant H3N2 viruses was measured by a microneutralization assay, with antibodies (**B**) continuously present in the medium throughout the experiment, or (**C**) removed at one-hour post-infection (hpi). MIC: minimal inhibitory concentration. Each bar represents the mean of two independent biological replicates. Each data point represents one replicate. **A–C** Strain names are abbreviated as follows: H3N2 A/Hong Kong/1/1968 (HK/1968), H3N2 A/Philippines/2/1982 (Philippines/1982), H3N2 A/Beijing/109/1992 (Beijing/1992), H3N2 A/Wuhan/359/1995 (Wuhan/1995), H3N2 A/Moscow/10/1999 (Moscow/1999), H3N2 A/New York/55/2004 (NY/2004), H3N2 A/Wisconsin/67/2005 (Wisconsin/2005), H3N2 A/Uruguay/716/2007 (Uruguay/2007), H3N2 A/Switzerland/9715293/2013 (Switzerland/2013), H3N2 A/Darwin/6/2021 (Darwin/2021), H3N8 A/mallard/Alberta/362/2017 (Alberta/2017), H4N6 A/mallard/Alberta/455/2015 (Alberta/2015), H7N3 A/Canada/rv444/2004 (Canada/2004), H7N9 A/Hong Kong/125/2017 (HK/2017), H7N9 A/Anhui/1/2013 (Anhui/2013), H14N5 A/mallard/Astrakhan/263/1982 (Astra/1982), and H15N2 A/Australian shelduck/Western Australia/1756/1983 (WAus/1983).

further substantiated that the antigenicity of the lower stem epitope in human H3N2 HA has changed over time due to the natural evolution of HA2 position 32.

## Discussion

Although the discovery of HA stem antibodies has motivated the development of broadly protective influenza vaccines, most known HA stem antibodies to date are specific to group 1 HA or cross-react with both group 1 and 2 HAs. By contrast, group 2-specific HA stem antibodies are not as well characterized. Through analyzing two non-competing group 2-specific HA stem antibodies, namely the lower stem antibody AG2-G02 and the central stem antibody 2F02, this study advances our understanding of broadly reactive antibody responses to influenza A viruses.

In the HA stem, the central stem epitope represents the most conserved epitope and is targeted by several known cross-group antibodies[11]. By contrast, the anchor epitope is mainly targeted by group 1-specific antibodies[45], whereas the lower stem epitope is only

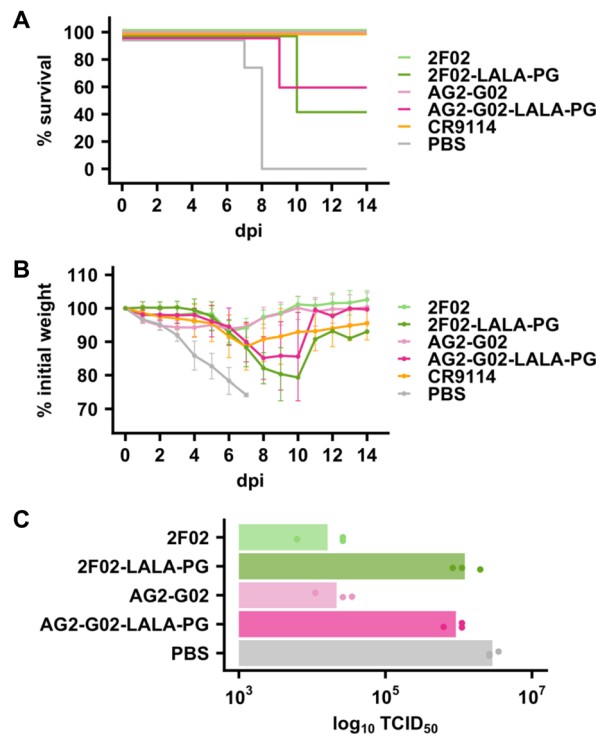

**Fig. 4 | In vivo protection activity of AG2-G02 and 2F02.** Female BALB/c mice at six weeks old were injected intraperitoneally with 5 mg/kg of the indicated antibody four hours prior to challenge with $5 \times LD_{50}$ of H3N2 Philippines/1982 (X-79). CR9114 was used as a positive control for protection. **A** Kaplan-Meier survival curves are presented ($n = 7$ for control, $n = 5$ for experimental groups). **B** The mean percentage of body weight change post-infection is shown ($n = 7$ for the control group, $n = 5$ for each experimental group). The humane endpoint was defined as a 25% weight loss from the initial weight on day 0. Data are presented as mean ± standard deviation. **C** Lung viral titers on day 3 post-infection are shown ($n = 3$ per group). Each bar represents the mean. Each data point represents one mouse.

known to be targeted by group 2-specific antibodies[19,28,29]. Since central stem antibodies have larger cross-reactivity breadth but weaker neutralization potency compared to lower stem antibodies[26], broadly protective influenza vaccines that simultaneously elicit high levels of both types would likely have superior protection efficacy and breadth to those that do not. At the same time, lower stem antibodies often compete with central stem antibodies, as shown by this study and others[37]. Besides, antibody responses are influenced by pre-existing immunity due to competition between circulating antibodies and B cell receptors[46–48]. Therefore, a high level of circulating antibodies targeting the lower stem epitope would most likely interfere with the B cell responses against the central stem epitope and vice versa. Nevertheless, this study showed that the competition between lower stem and central stem antibodies is not universal, as exemplified by the lack of competition between AG2-G02 and 2F02 or FI6v3. This observation suggests the possibility of minimizing the interference between antibody responses targeting lower stem and central stem epitopes, which should be explored in the future design of broadly protective influenza vaccines.

A major highlight in this study is identifying the altered antigenicity of the lower stem epitope in the human H3N2 virus due to the natural evolution of HA2 position 32. Similarly, a recent study has also suggested that sequence differences at HA2 position 32 across different group 2 HA subtypes restrict the cross-reactivity of lower stem antibodies[26]. In fact, mutations at this position have been known to reduce the binding affinity of one of the classic lower stem antibodies, CR8043[26,29], as well as 9H10[49], which is a murine lower stem antibody.

Furthermore, a deep mutational scanning study has shown that HA2 position 32 has high mutation tolerance in vitro[50], suggesting minimal fitness costs for altering the antigenicity of the lower stem epitope, although further characterization in vivo is needed. At the same time, lower stem antibodies have higher neutralization potency than central stem antibodies[26]. Therefore, while the lower stem epitope remains an attractive target for the development of broadly protective influenza vaccines[26], potential resistance mutations as well as natural sequence variations at HA2 position 32 need to be considered.

Ancestral human H3N2 strains carry $T32_{HA2}$, whereas more recent ones carry $R32_{HA2}$. Consistently, we showed that people who were born after 2011 have weaker pre-existing immunity against the HA stem of H3N2 strains with $T32_{HA2}$. By contrast, such pre-existing immunity was stronger in older people who were born when $T32_{HA2}$ was prevalent in human H3N2 strains. Since most avian H3 strains have $T32_{HA2}$ (Fig. S9A), our observation suggested that older people could generate a better recall B cell response against the HA stem of zoonotic H3 strains. Consistently, our preliminary results indicated that plasma samples from older people had better cross-reactivity than those from young people to an avian H3 HA (Fig. S9B), although head antibodies may also contribute to such a difference. As human H3N2 strains with $R32_{HA2}$ continue the circulate, we anticipate that pre-existing immunity to the lower stem epitope of zoonotic H3 strains would diminish over time at the population level. This may potentially increase the public health risks of spillover of zoonotic H3 strains, provided that the lower stem epitope is a major cross-neutralizing epitope in the H3 subtype[26].

Multiple broadly protective influenza vaccine candidates that are designed based on group 1 HA stem have been shown to elicit antibody responses against group 1 HAs but not group 2 HAs[20–22]. On the contrary, a recently completed phase I clinical trial of an H10 stem-based vaccine has demonstrated its ability to induce antibody responses against group 2 HAs (ClinicalTrials.gov identifier: NCT04579250)[26]. These observations indicate the complementary of group 1 and group 2 HA stem-based influenza vaccines. Nevertheless, our previous study has suggested that resistance mutations against HA stem antibodies are much easier to arise in H3 HA than in H1 HA[4,51], indicating that the development of broadly protective influenza vaccines against group 2 HA might be more challenging than for those against group 1 HA. Given the importance of antibody characterization in vaccine development, an optimal design of group 2 HA stem-based vaccines would benefit from continued study of group 2 HA stem antibodies in the future.

## Methods
### Sample collection
Plasma samples were collected from elderly volunteers (ages 59–65) and young volunteers (ages 17–25) between January and March 2020 at the Red Cross in Hong Kong. Plasma samples from infants were collected between January and March 2022 in Guangzhou, China. These samples were obtained from ethylenediaminetetraacetic acid (EDTA)-anticoagulated peripheral blood. Peripheral blood samples were centrifuged at $3000 \times g$ for 10 min at room temperature to isolate plasma, which was then stored at −80 °C until needed. Plasma samples from adults that were collected before 2003 were purchased from Bio-Collections Worldwide Inc.

### Cell lines
MDCK-SIAT1 cells (Madin-Darby canine kidney cells engineered for stable human 2,6-sialyltransferase expression,(Sigma-Aldrich, Cat. No. 05071502) and HEK 293 T cells (human embryonic kidney cells, (ATCC, Cat. No. CRL-3216) were grown at 37 °C with 5% $CO_2$ in Dulbecco's modified Eagle's medium (DMEM) enriched with high glucose (Thermo Fisher Scientific), along with 10% heat-inactivated fetal bovine serum (FBS, Thermo Fisher Scientific), 1× penicillin-streptomycin at final

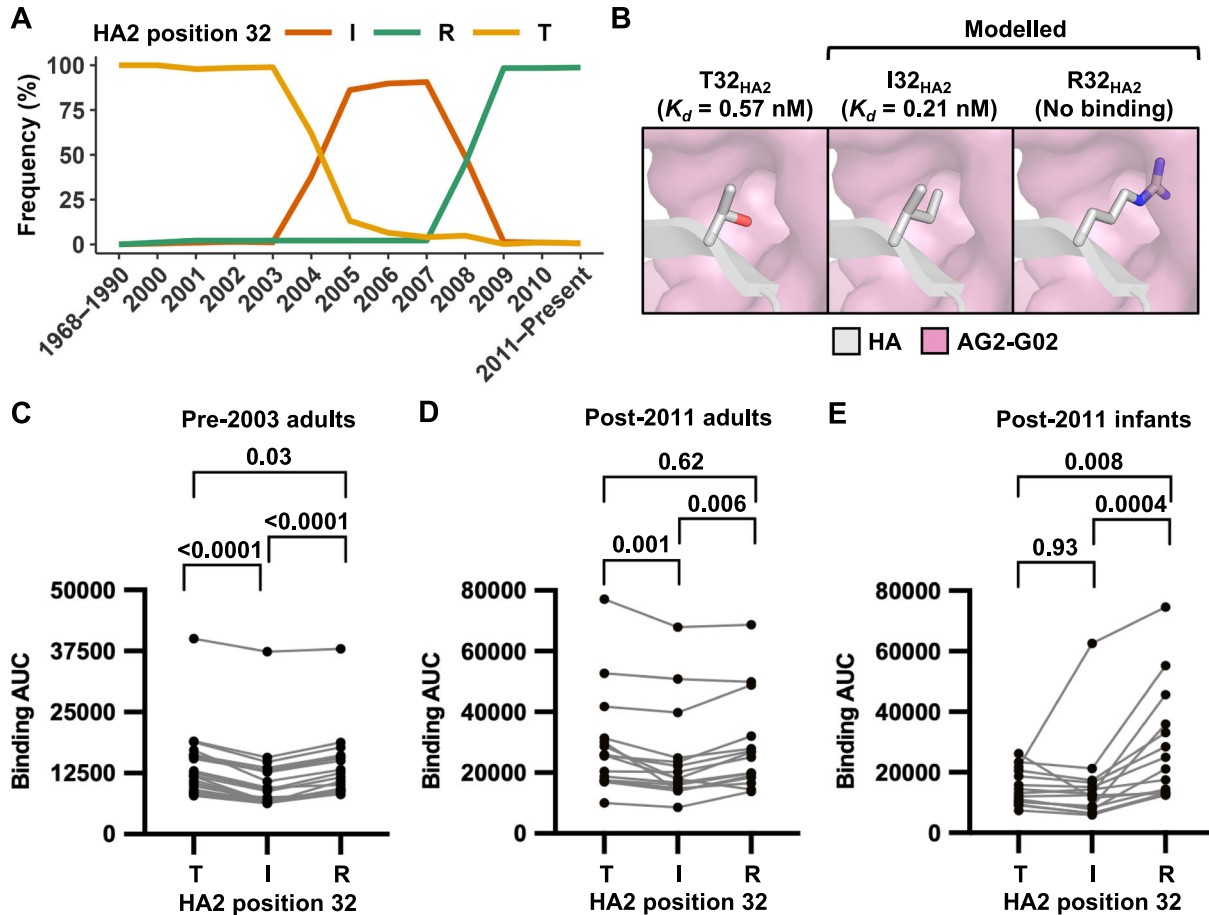

**Fig. 5 | Evolution of HA2 position 32 influences the antigenicity of the lower stem epitope. A** Occurrence of different amino acid variants at HA2 position 32 of human H3N2 strains from 1968 to the present was analyzed based on 33,000 human H3N2 HA sequences from the GSAID database[65]. The x-axis represents the years, while the y-axis shows the percentage of strains containing a particular amino acid at HA2 position 32 in a given year. Thr, Ile, and Arg are represented by orange, brown, and green lines, respectively. **B** The structures of AG2-G02 binding to HA stem with $I32_{HA2}$ and $R32_{HA2}$ are modeled. The binding affinity ($K_d$) of AG2-G02 Fab to different HA stem variants is indicated in parentheses. **C–E** The binding of plasma samples from (**C**) pre-2003 adults ($n = 20$), (**D**) post-2011 adults ($n = 20$), and (**E**) post-2011 infants ($n = 15$) to H3 stem with either $T32_{HA2}$, $I32_{HA2}$ or $R32_{HA2}$ is measured by ELISA. The y-axis represents the area under the curve (AUC) of four serial 10-fold dilutions of serum (1:100, 1:1,000, 1:10,000, and 1:100,000). *P*-values were determined using a paired two-tailed Student's *t*-test.

concentrations of 100 U mL$^{-1}$ of penicillin and 100 µg mL$^{-1}$ of streptomycin (Thermo Fisher Scientific), as well as 1× GlutaMax (Thermo Fisher Scientific). MDCK-SIAT1 and HEK 293 T cells were passaged every 3–4 days using a 0.25% trypsin-ethylenediaminetetraacetic acid (EDTA) solution (Thermo Fisher Scientific). Expi293F cells (human embryonic kidney cells, Thermo Fisher Scientific, Cat. No. A14527) were cultured at 37 °C with 8% $CO_2$ in Expi293 Expression Medium (Thermo Fisher Scientific). Sf9 cells (*Spodoptera frugiperda* ovarian cells, ATCC, Cat. No. CRL-1711) were grown in Sf-900 II SFM medium (Thermo Fisher Scientific).

### Mice
Six-week-old female BALB/c mice (Jackson Laboratory) were used for all animal experiments, with seven animals in the control group and five animals in the experimental group.

### Influenza virus
The recombinant influenza H3N2 A/Darwin/6/2021 virus and H3N2 A/Moscow/10/1999 virus were generated based on the A/PR/8/34 (PR8) eight-plasmid reverse genetic system[52]. The PR8 backbone was used to generate 7:1 reassortants, with the entire HA coding region being replaced by those from H3N2 strains. Transfection was performed in HEK 293 T/MDCK-SIAT1 cells co-culture (ratio of 6:1) at 60% confluence using Lipofectamine 2000 (Life Technologies) according to the

manufacturer's instructions. At 24 h post-transfection, cells were washed twice with phosphate-buffered saline (PBS), and cell culture medium was replaced with OPTI-MEM medium supplemented with 1 µg mL$^{-1}$ tosyl phenylalanyl chloromethyl ketone (TPCK)-trypsin. The virus was harvested at 72 h post-transfection. $TCID_{50}$ (tissue culture infectious dose 50) was used to quantify the viral titer. Briefly, serial dilutions of the virus sample were made and added to MDCK-SIAT1 cells supplemented with 1 µg mL$^{-1}$ TPCK-trypsin. After incubation for 72 h, the cells were examined for cytopathic effect (CPE). The dilution that caused infection in 50% of the wells was used to calculate the $TCID_{50}$. The Reed-Muench method was used to estimate the viral titer in terms of $TCID_{50}$ mL$^{-1}$. The following strains of influenza virus were obtained from BEI Resources (https://www.beiresources.org/): H3N2 A/Philippines/2/1982 (cat #: NR-28649), H3N2 A/Wisconsin/67/2005 (cat #: NR-41800). Mouse-adapted H3N2 A/Philippines/2/1982 (X-79, 6:2 A/PR/8/34 reassortant) virus was grown in 10-day-old embryonated chicken eggs at 37 °C for 48 h and were cooled at 4 °C overnight. Cell debris was removed by centrifugation at 4000 × *g* for 20 min at 4 °C.

### HA proteins
The H1 and H3 stems as well as their mutants, along with the HAs from H3N2 A/Darwin/6/2021, H3N8 A/mallard/Alberta/362/2017 and H4N6 A/mallard/Alberta/455/2015 HA, were produced in the baculovirus expression system in the lab. HA proteins from the following strains

were obtained from BEI Resources (https://www.beiresources.org/): H3N2 A/New York/55/2004 (cat #: NR-19241), H3N2 A/Uruguay/716/2007 (cat #: NR-15168), H7N9 A/Hong Kong/125/2017 (cat #: NR-51367), H7N9 A/Anhui/1/2013 (cat #: NR-44081), H7N3 A/Canada/rv444/2004 (cat #: NR-43740). HA proteins of the following strains were purchased from SinoBiological: H3N2 A/HongKong/1/1968, H3N2 A/Beijing/109/1992, H3N2 A/Wuhan/359/1995, H3N2 A/Moscow/10/1999, H3N2 A/Switzerland/9715293/2013, H14N5 A/mallard/Astrakhan/263/1982, and H15N2 A/Australian shelduck/Western Australia/1756/1983.

## Expression and purification of Fabs and IgGs
Plasmids encoding the heavy and light chains of IgG and Fabs were generated by cloning into the phCMV3 vector with a mouse kappa signal peptide. These plasmids were transfected into Expi293F cells using the ExpiFectamine 293 transfection kit (Gibco) in a 2:1 mass ratio of heavy to light chains, following the manufacturer's protocol. Cells were cultured at 37 °C, 8% $CO_2$, and 125 rpm. After six days, the supernatant was harvested and centrifuged at 4000 × $g$ for 30 min at 4 °C. Purification was performed by adding CaptureSelect CH1-XL beads (Thermo Fisher Scientific) to the supernatant, followed by overnight shaking at 4 °C. The beads were eluted with 60 mM sodium acetate at pH 4.0, neutralized with 1 M Tris at pH 8.0, and then buffer-exchanged three times to Phosphate-Buffered Saline (PBS) using 50 kDa molecular weight cutoff (MWCO) concentrators (Millipore) for IgGs and 30 kDa MWCO concentrators for Fabs. The antibody concentrations were determined using a Nanodrop One (Thermo Fisher Scientific) at 280 nm, and the antibodies were stored at 4 °C.

## Expression and purification of HA recombinant proteins
The H1[18] and H3[53] stem constructs, along with the H3N2 A/Darwin/6/2021 HA, H4N6 A/mallard/Alberta/455/2015 HA, and H3N8 A/mallard/Alberta/362/2017 HA ectodomain, were cloned into a customized baculovirus transfer vector. Each construct included an N-terminal gp67 signal peptide, a BirA biotinylation site, a thrombin cleavage site, a trimerization domain, and a C-terminal 6×His-tag. Recombinant bacmid DNA was created using the Bac-to-Bac system (Thermo Fisher Scientific) following the manufacturer's instructions. Baculovirus was generated by transfecting the purified bacmid DNA into adherent Sf9 cells with Cellfectin reagent (Thermo Fisher Scientific) according to the manufacturer's protocol. The baculovirus was then amplified by passaging in adherent Sf9 cells at a multiplicity of infection (MOI) of 1. Recombinant HA stem constructs were expressed in 1 L of suspension Sf9 cells at an MOI of 1. Three days post-infection, the cells were harvested by centrifugation at 4000 × $g$ for 25 min. Soluble His-tagged recombinant ectodomains of H3N2 A/Darwin/6/2021 HA, H4N6 A/mallard/Alberta/455/2015 HA, H3N8 A/mallard/Alberta/362/2017 HA, as well as H1 and H3 stem constructs, were purified from the supernatant using Ni Sepharose Excel resin (Cytiva) for affinity chromatography. The resin was washed with two column volumes each of wash buffer 1 (20 mM sodium phosphate, 500 mM NaCl, 20 mM imidazole, pH 7.4) and wash buffer 2 (20 mM sodium phosphate, 500 mM NaCl, 40 mM imidazole, pH 7.4) to remove non-specific proteins. The proteins were then eluted using a high-imidazole buffer (20 mM sodium phosphate, 500 mM NaCl, 200 mM imidazole, pH 7.4) following a 10-min incubation. Subsequently, size exclusion chromatography was performed on a HiLoad 16/100 Superdex 200 prep grade column (Cytiva) in 20 mM Tris-HCl (pH 8.0) with 100 mM NaCl. The purified protein was concentrated using an Amicon spin filter (Millipore Sigma) and filtered through a 0.22 µm centrifuge Tube Filter (Costar). Protein concentration was quantified by measuring $A_{280}$ absorbance using a Nanodrop One (Thermo Fisher Scientific).

## ELISA for human plasma samples
Assays were performed using Nunc MaxiSorp plates (Thermo Fisher Scientific), which were coated overnight with 100 ng per well of recombinant HA protein in phosphate-buffered saline (PBS). The following day, plates were washed three times with PBS containing 0.1% Tween 20. The plates were then blocked with 100 µL of Chonblock blocking/sample dilution ELISA buffer (Chondrex Inc., Redmond, US) and incubated at room temperature for 1 h. For human plasma samples, each sample was diluted to an initial concentration of 1:100, followed by ten-fold dilutions (1:1000, 1:10,000, 1:100,000) prepared in Chonblock blocking/sample dilution ELISA buffer and added to the antigen wells. Plates were incubated for 2 h at 37 °C. After washing the plates three times with PBS containing 0.1% Tween 20, each well was incubated with 100 µL of anti-human IgG secondary antibody (goat anti-human IgG, HRP-conjugated; Thermo Fisher Scientific, Cat. #31410) for 1 h at 37 °C. After six washes with 1× PBS containing 0.05% Tween 20, 100 µL of 1-Step Tetramethylbenzidine (TMB) ELISA Substrate Solution (Thermo Fisher Scientific) was added to each well. After incubation for 10 min, the reaction was stopped with 50 µL of 2 M $H_2SO_4$ solution, and absorbance values were measured at 450 nm using a BioTek Synergy HTX Multimode Reader (Agilent).

## Monoclonal antibody binding using ELISA
Assays used Nunc MaxiSorp plates (Thermo Fisher Scientific) coated overnight with 100 ng recombinant HA protein in phosphate-buffered saline (PBS). The next day, plates were washed three times with PBS and 0.1% Tween 20. Blocking was carried out with 5% nonfat dry milk for 2 h at room temperature. An initial concentration of 100 µg mL$^{-1}$ was used to calculate the area under the curve (AUC), and 10-fold dilutions were done. For endpoint ELISA experiments, the purified recombinant antibodies were diluted to 10 µg mL$^{-1}$ in 5% Nonfat dry milk, added to the plates, and incubated for 2 h at 37 °C. Following three washes with PBS containing 0.1% Tween 20, each well was incubated with 100 µL of anti-human IgG secondary antibody (goat anti-human IgG, HRP-conjugated; Thermo Fisher Scientific, Cat. #31410) for one hour at 37 °C. After six washes with 1× PBS containing 0.05% Tween 20, 100 µL of 1-Step TMB ELISA Substrate Solution (Thermo Fisher Scientific) was added to each well. After a 10-min incubation, the reaction was stopped with 50 µL of 2 M $H_2SO_4$ solution, and absorbance values were measured at 450 nm using a BioTek Synergy HTX Multimode Reader (Agilent).

## Biolayer interferometry binding assay
Binding assays were performed using biolayer interferometry (BLI) on an Octet RED96e instrument (Sartorius). For the measurement of $K_d$, His-tagged HA protein (20 µg mL$^{-1}$) in 1× kinetics buffer (1× PBS, pH 7.4, with 0.002% Tween 20) was loaded onto HIS1K biosensors and incubated with Fabs at concentrations of 300 nM, 100 nM, and 33.3 nM. The assay consisted of five steps: baseline (60 s in 1× kinetics buffer), loading (60 s with His-tagged HA protein), a second baseline (60 s in 1× kinetics buffer), association (60 s with Fabs), and dissociation (60 s in 1× kinetics buffer). $K_d$ values were estimated using a 1:1 binding model.

For competition assays, the H3 stem was first loaded onto HIS1K biosensors for 120 s. Antibody binding was measured by exposing the sensors to 100 nM of the first antibody in 1× kinetics buffer for 120 s. The degree of additional binding was then assessed by exposing the sensors to 100 nM of a second antibody in the presence of the first antibody (100 nM) for another 120 s.

The competition index (CI) for an antibody Ab1 competing with a pre-bound antibody Ab2 was calculated as:

$$CI = \frac{\left(max\ response_{with\ Ab2\ pre-bound}\right) - \left(max\ response_{without\ Ab2\ pre-bound}\right)}{max\ response_{without\ Ab2\ pre-bound}}$$

where "$max\ response_{without\ Ab2\ pre-bound}$" is the maximum response of Ab1 in the absence of Ab2 pre-bound and "$max\ response_{with\ Ab2\ pre-bound}$" is the maximum response of Ab1 in the presence of Ab2 pre-bound.

## Cryo-EM sample preparation and data collection

The purified H3N8 ectodomain, uncleaved protein, was mixed with each Fab at a 1:4 molar ratio and incubated at 4 degrees overnight before size exclusion chromatography. The peak fraction of the Fab-HA complex was eluted in 20 mM Tris-HCl, pH 8.0, and 100 mM NaCl and concentrated to around 3 mg mL$^{-1}$, and mixed with n-octyl-β-D-glucoside (Anagrade) at a final concentration of 0.1% w/v for cryo-EM sample preparation. Cryo-EM grids were prepared using a Vitrobot Mark IV machine. An aliquot of 3 μL sample was applied to a 300-mesh Quantifoil R1.2/1.3 Cu grid pre-treated with glow-discharge. Excess liquid was blotted away using filter paper with blotting force 0 and blotting time 3 s. The grid was plunge-frozen in liquid ethane. Data collection was done on a Titan Krios microscope equipped with a Gatan detector. Images were recorded at 81,000× magnification, corresponding to a pixel size of 0.53 Å/pix at the super-resolution mode of the camera. A defocus range of −0.8 μm to −3 μm was used with a total dose of 57.35 e-/Å$^2$.

## Cryo-EM data processing

Data processing was conducted using CryoSPARC Live (v4.5)[54]. Movies were subjected to motion correction and contrast transfer function (CTF) estimation, and particles were picked with the CryoSPARC blob picker followed by 2D classification. The best classes identified by the blob picker served as templates for CryoSPARC template pickers. The resulting particles underwent multiple rounds of 2D classification to ensure thorough cleanup before proceeding to ab initio reconstruction. The most effective class from the ab initio reconstruction was then subjected to homogeneous refinement, reference-based motion correction, followed by an additional round of homogeneous refinement, local and global CTF estimation, and non-uniform refinement. Finally, the map was sharpened using DeepEMhancer (v0.15)[55].

## Low resolution cryo-EM analysis

For the H3N8 complex with AG2-G02 and 2F02 Fabs, data were collected using a Glacios 2 Cryo-TEM, equipped with a Falcon 4i Direct Electron Detector. Images were recorded at a magnification of 150,000×, with a pixel size of 0.96 Å/pix. A defocus range of −0.3 μm to −3 μm was applied, along with a total dose of 60 e-/Å$^2$. Data processing was conducted using CryoSPARC Live (v4.5)[54] as described above.

## Model building and refinement

An initial model for the cryo-EM maps was built using ModelAngelo (v0.3)[56], an automated atomic model-building program. This model was then fitted into the cryo-EM density map using UCSF Chimera (v1.16)[57], followed by manual adjustments in Coot (v0.9.8.1)[58] and refinement with the Phenix real-space refinement program (v1.20)[59]. Refinement was repeated iteratively until no further significant improvements were observed.

## Sequence conservation analysis

Full-length human H3N2 HA and avian H3 HA protein sequences from different subtypes were downloaded from the Global Initiative for Sharing Avian Influenza Data (GISAID; https://gisaid.org). Sequences corresponding to HA2 were extracted. We then calculated the percentage of each character at the HA2 position 32 across the strains per year. The percentages were written to a CSV file to generate a graph depicting the frequencies of each amino acid variant over time.

## Structural analysis of HA-antibody complexes

Buried surface areas upon binding and paratope residues of AG2-G02, 2F02, CR9114 (PDB 4FQY)[37], FI6v3 (PDB 3ZTJ)[32], CR8020 (PDB 3SDY)[28], CR8043 (PDB 4NM8)[29], and ADI-85666 (PDB 9BDF)[38] were analyzed using PDBePISA[60]. The CDR regions and germline gene usage of each antibody were annotated using IgBLAST[61]. The molecular interactions of AG2-G02 and 2F02 in complex with H3N8 HA were analyzed and visualized using PyMOL (Schrödinger v2.5).

## Microneutralization assay

For the microneutralization assay, MDCK-SIAT1 cells were seeded in 96-well plates. After reaching 100% confluency, MDCK-SIAT1 cells were washed once with 1× PBS. Minimal essential media (Gibco) containing 25 mM 4-(2-hydroxyethyl)−1-piperazineethanesulfonic acid (HEPES, Gibco) was then added to the cells. Monoclonal antibodies were serially diluted 2-fold starting from 100 μg mL$^{-1}$ and mixed with 100 TCID$_{50}$ (median tissue culture infectious dose) of viruses at equal volume and incubated at 37 °C for 1 h. Subsequently, the mixture was inoculated into cells and incubated at 37 °C for another hour. Supernatants were replaced with minimal essential media containing 25 mM HEPES and 1 μg/mL TPCK-trypsin. In some experiments, as indicated, antibodies were also added back to the supernatants at the same concentrations before medium replacement. Notably, it is a common practice to maintain the antibodies in the medium for the entire duration of the microneutralization assay to test the neutralization activity of HA stem antibodies[28,29,62]. Plates were incubated at 37 °C for 48 h, and virus presence was assessed by cytopathic effect to determine the minimum inhibitory concentration.

## HAI assay

Briefly, 50 μl of H3N2 A/Darwin/6/2021 virus was mixed with 2-fold serial dilutions of plasma samples and incubated for 30 min. After incubation, 50 μl of 1% turkey red blood cells were added to the wells. The highest dilution of the serum that prevented hemagglutination was recorded, and the HI titer was calculated.

## ADCC reporter assay

A total of $1.5 \times 10^4$ MDCK cells were plated in white, flat-bottom 96-well plates and incubated overnight at 37 °C. The following day, cells were washed with PBS and exposed to $1.5 \times 10^6$ PFU/mL of H3N2 A/Philippines/2/1982 virus for 24 h. After infection, the medium was removed and wells were treated with serial dilutions of antibodies (1:10 in RPMI 1640), together with effector Jurkat cells expressing human FcγRIIIa V158 and an NFAT-driven luciferase reporter. Plates were incubated for 6 h at 37 °C, after which Bio-Glo luciferase reagent (Promega) was added. Luminescence was recorded after 10 min in the dark using a BioTek Synergy HTX Multimode Reader (Agilent). Data were analyzed in Prism, and area under the curve (AUC) values were determined.

## Flow cytometry analysis

MDCK-SIAT1 cells were seeded in 12-well plates. At 100% confluency, cells were washed with 1× PBS and infected with H3N2 A/Darwin/6/2021 virus and H3N2 A/Moscow/10/1999 virus at an MOI of 0.1 in minimal essential media (Gibco) containing 25 mM HEPES, and 1 μg mL$^{-1}$ TPCK-trypsin (Sigma). At 36 h post-infection, cells were fixed with 4% paraformaldehyde overnight at 4 °C. On the next day, paraformaldehyde was removed, and the cells were blocked in blocking buffer (1× PBS with 2% FBS and 0.1% BSA) for 30 min. Cells were incubated with 20 μg mL$^{-1}$ of the indicated antibody in blocking buffer at 4 °C for 1 h. Subsequently, the cells were washed twice with blocking buffer and incubated with 1:500 dilution of PE anti-human IgG Fc (clone M1310G05, BioLegend, catalog #: 410708) at 4 °C for 1 h. The cells were then washed thrice and resuspended in 1× PBS for flow cytometry analysis using a FACSymphony A1 (BD Biosciences). Data were analyzed using FlowJo v10.10 Software (BD Life Sciences).

## Prophylactic protection experiments

Mice were housed in a specific-pathogen-free (SPF) BSL-2 facility under a 12-h light/dark cycle at 18−22 °C with 40−60% relative humidity, and were provided food and water ad libitum. Six-week-old female BALB/c

mice (Jackson Laboratory) ($n = 5$–7) were anesthetized with isoflurane and intranasally infected with 5× lethal dose ($LD_{50}$) of H3N2 A/Philippines/2/1982 (X-79, 6:2 A/PR/8/34 reassortant, mouse-adapted) virus. Mice received the indicated antibodies at 5 mg/kg intraperitoneally 4 h prior to infection. Weight loss was monitored daily for 14 days. The humane endpoint was defined as a weight loss of 25% from the initial weight at day 0. While our BALB/c mice were not modified to facilitate interaction with human IgG1, human IgG1 could interact with mouse Fc gamma receptor[63,64]. To determine the lung viral titers at day 3 post-infection, the lungs of infected mice were harvested and homogenized in 1 mL of minimal essential media using a gentleMACS C Tube (Miltenyi Biotec). Subsequently, virus titers were measured by the $TCID_{50}$ assay.

## Ethical statement

The study received approval from the Human Research Ethics Committee at Guangdong Women and Children Hospital (Approval number: 202101231) and The Chinese University of Hong Kong (IRB: 2020.229). Our research conformed to all ethical regulations, with approval from the Human Research Ethics Committee at Guangdong Women and Children Hospital and the Chinese University of Hong Kong. All study procedures were performed on volunteers after verbal informed consent. For infant participants, verbal consent was obtained from the parents or guardians. Animal experiments were conducted in a BSL-2 facility in accordance with protocols approved by UIUC Institutional Animal Care and Use Committee (IACUC, protocol number: 22215).

## Reporting summary

Further information on research design is available in the Nature Portfolio Reporting Summary linked to this article.

## Data availability

Cryo-EM maps have been deposited to the Electron Microscopy Data Bank under accession codes: EMD-48873 and EMD-48874. The refined models have been deposited in the RCSB Protein Data Bank under accession codes 9N4E and 9N4F. Structures from the following identifiers from the Protein Data Bank (PDB) were used in this study: 4FQY, 3ZTJ, 3SDY, 4NM8, and 9BDF. Sequences used for the analysis of influenza virus evolution were downloaded from GISAID. Source data are provided with this paper.

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

## Acknowledgments

We thank Kristen Flatt and the Materials Research Laboratory Central Research Facilities at University of Illinois Urbana-Champaign, as well as Frank Vago and the Cryo-EM Facility at Purdue University, for access to cryo-EM instrumentation and data collection. We also thank Disha Bhavsar and Florian Krammer for providing the H3N2 A/Philippines/2/1982 (X-79) virus, and Florian Krammer for helpful comments on the manuscript. This work is supported by the Carl R. Woese Institute for Genomic Biology Postdoctoral Fellowship (H.L.), UIUC William T. and Lynn Jackson Graduate Student Fellowship (W.O.O.), Emergency Key Program of Guangzhou Laboratory, China [Grant No. EKPG22-30-6] (C.K.P.M), Health and Medical Research Fund (no. 24230352) (C.K.P.M), Research Grants Council of the Hong Kong Special Administrative Region, China (HKU C7053-24G) (C.K.P.M), Searle Scholars Program (N.C.W.), Vallee Scholars Program (N.C.W.), and Howard Hughes Medical Institute Emerging Pathogens Initiative (N.C.W.).

## Author contributions

A.B.G., H.L., W.O.O., and N.C.W. conceived and designed the study. A.B.G., H.L., T.P., Q.W.T., and W.O.O. performed the experiments. Y.L., M.L., C.K.P.M., Y.S.T., and H.L. collected the human samples. A.B.G., H.L., and N.C.W. wrote the paper, and all authors reviewed and edited the paper.

## Competing interests

N.C.W. consults for HeliXon. The authors declare no other competing interests.
