## [Peer Review file · Nature Communications]

Characterization of two non-competing antibodies to influenza H3N2 hemagglutinin stem reveals its evolving antigenicity

Corresponding Author: Dr Nicholas Wu

Version 0:

Reviewer comments:

Reviewer #1

(Remarks to the Author)

Here, Gopal and colleagues describe the use of machine learning method to identify broadly binding antibodies against the influenza A virus hemagglutinin. Two of these mAbs, namely AG2-G02 and 2F02 bind to H3 at discreet regions of the stalk, the more structurally/antigenically conserved region of the typically highly variable HA. The notable finding of the study demonstrates that both antibodies are capable of binding to the HA stalk non-competitively. However, there are several noted weaknesses that detracts the present study in impacting the field.

The binding and the neutralizing profiles are not particularly extraordinary compared to what have been previously reported. In fact, AG2-G02 and 2F02 are arguably inferior to previously characterized ones. The lack of neutralizing capacity in vitro is notable since their epitopes are not all different from ones we current know. While we do not expect that these two mAbs to behave identically to others, it perhaps reflects that not all stalk antibodies are equal. With that said, the neutralization assays as describe may not be the best protocol to determine the neutralizing activity of antibodies that target the stalk. In Figure 3, it is noteworthy that even CR8020 (and CR8043) appear to have suboptimal IC50. Given their mechanism of action, stalk antibodies are also included in the media (during the 48 hours of incubation) or the agar overlay (when performing a plaque reduction neutralization assay). While heatmaps are pretty to look at, it's difficult to determine what the actual IC50 are for mAbs. Please reconsider using another way to show IC50.

While the authors do mention and show that both mAbs protect against lethal challenge in vivo with minimal or no in vitro neutralizing activity – it would make the study stronger if the authors would make murine versions of the 2 mAbs to demonstrate that protection is mediated by Fc-dependent immunity. This would increase the impact of the study since it appears that even with weak or no neutralizing activity, both mAbs appear to work better than CR9114.

Reviewer #2

(Remarks to the Author)

In this manuscript, Akshita B. Gopal. et al, characterized two group 2 HA stem antibodies AG2-G02 and 2F02. Both antibodies had minimal neutralization activity in vitro but could offer prophylactic protection in vivo. Cryo-EM structures revealed the binding details, and AG2-G02 binds to the lower stem epitope while 2F02 binds to the central stem epitope. Further sequence analysis showed that the natural evolution of HA2 position 32 restricted the binding of AG2-G02 to recent human H3N2 HAs and influences the binding of human plasma samples. Overall, these findings advance our understanding of the antigenicity of HA stem, and the manuscript is well written, and data interpretation is matching the conclusion. However, there are several concerns regarding the presented data before publication.

1. Both antibodies have minimal neutralization activity in vitro, but could offer protection in vivo, additional experiments, such as ADCC, are needed to reveal the protection mechanism of these two antibodies.
2. Please give an explanation why 2F02 does not have minimal neutralization activity, compared to previously reported neutralizing antibodies targeting the central stem.
3. The authors should further clarify the important implications of their findings for the development of broadly protective

influenza vaccines.

4. Female BALB/c mice at 6 weeks old (n = 4-5 per group) were used for the prophylactic protection experiments. In ordinary experiments, 5 or more mice were needed, and the small number of mice might affect the accuracy of the results.

5. In the methods section, "The purified H3N8 full-length, uncleaved protein" should be "The purified H3N8 ectodomain, uncleaved protein".

6. The purification process of HA ectodomain proteins is not clearly mentioned.

Version 1:

Reviewer comments:

Reviewer #1

(Remarks to the Author)

Authors have addressed the concerns previously raised.

Reviewer #2

(Remarks to the Author)

I am pleased to confirm that all of my concerns have been satisfactorily addressed. I have no further comments.

Reviewer #1

Here, Gopal and colleagues describe the use of machine learning method to identify broadly binding antibodies against the influenza A virus hemagglutinin. Two of these mAbs, namely AG2-G02 and 2F02 bind to H3 at discreet regions of the stalk, the more structurally/antigenically conserved region of the typically highly variable HA. The notable finding of the study demonstrates that both antibodies are capable of binding to the HA stalk non-competitively. However, there are several noted weaknesses that detracts from the present study in impacting the field.

Response: Thank you for the encouraging and constructive comments.

Major concerns:

1. The binding and the neutralizing profiles are not particularly extraordinary compared to what have been previously reported. In fact, AG2-G02 and 2F02 are arguably inferior to previously characterized ones. The lack of neutralizing capacity in vitro is notable since their epitopes are not all different from ones we current know. While we do not expect that these two mAbs to behave identically to others, it perhaps reflects that not all stalk antibodies are equal. With that said, the neutralization assays as describe may not be the best protocol to determine the neutralizing activity of antibodies that target the stalk.

Response: As described in the Methods section of our first submission, our previous neutralization assay was performed as follows:

“Monoclonal antibodies were serially diluted 10-fold starting from 100 $\mu\text{g mL}^{-1}$ and mixed with 100 TCID₅₀ (median tissue culture infectious dose) of viruses at equal volume and incubated at 37°C for 1 hour. Subsequently, the mixture was inoculated into cells and incubated at 37°C for another hour. Cell supernatants were discarded and replaced with minimal essential media containing 25 mM HEPES, and 1 $\mu\text{g mL}^{-1}$ TPCK-trypsin (Sigma). Plates were incubated at 37°C for 48 hours, and virus presence was detected by monitoring the cytopathic effects (CPE) ...”

While this protocol removed the antibodies after one hour of infection, we realize that this practice differs from a published protocol for neutralization assays for HA stem antibodies, which keep the antibodies in the medium for the entire duration of the microneutralization assay (PMID: 19079604, PMID: 21737702, PMID: 24335589). Therefore, we have performed another neutralization experiment following the published protocol. The results are presented in Figure 3B-C and described in the Results section:

Lines 161-169: “We further assessed the neutralization activity of AG2-G02 and 2F02. While 2F02 demonstrated measurable neutralization activity against all five tested H3 strains spanning from 1982 to 2021 (Figure 3B), AG2-G02 only neutralized four and not H3N2 A/Darwin/6/2021, consistent with its lack of binding to this strain (Figure 3A-B). Notably, the neutralization activity of AG2-G02, 2F02, and our positive control antibodies CR8020 and CR8043 was much weaker when they were not maintained in the medium throughout the assay (Figure 3C), consistent with the known ability of HA antibodies in inhibiting viral release³⁹⁻⁴³. Together, these results demonstrated that both AG2-G02 and 2F02 are neutralizing antibodies with broad reactivity against H3 HAs.”

We have also updated the Methods section accordingly:

Lines 525-529: “In some experiments, as indicated, antibodies were also added back to the supernatants at the same concentrations before medium replacement. Notably, it is a common practice to maintain the antibodies in the medium for the entire duration of the microneutralization assay to test the neutralization activity of HA stem antibodies^{29,63,64}.”

2. In Figure 3, it is noteworthy that even CR8020 (and CR8043) appear to have suboptimal IC50. Given their mechanism of action, stalk antibodies are also included in the media (during the 48 hours of incubation) or the agar overlay (when performing a plaque reduction neutralization assay). While heatmaps are pretty to look at, it's difficult to determine what the actual IC50 are for mAbs. Please reconsider using another way to show IC50.

Response: In the revised manuscript, we replaced the heatmaps with bar graphs to present the neutralization data for a clearer comparison. Also, our initial manuscript presented the neutralization data as IC50, but they were in fact minimum inhibitory concentration (MIC). This has been corrected in the revised manuscript.

3. While the authors do mention and show that both mAbs protect against lethal challenge in vivo with minimal or no in vitro neutralizing activity – it would make the study stronger if the authors would make murine versions of the 2 mAbs to demonstrate that protection is mediated by Fc-dependent immunity. This would increase the impact of the study since it appears that even with weak or no neutralizing activity, both mAbs appear to work better than CR9114.

Response: In the revised manuscript, we have included additional data to illustrate the importance of Fc-mediated effector functions in the protection activity of AG2-G02 and 2F02. These new results are shown in Figure 4, Figure S4, and Figure S5, and described in the Results section:

Lines 176-186: “To investigate the role of Fc-mediated effector functions in the in vivo protection activity of AG2-G02 and 2F02, a LALA-PG variant was generated for each antibody to eliminate their Fc-mediated effector functions⁴⁴. Mice treated with the LALA-PG variants of both AG2-G02 and 2F02 exhibited reduced survival (Figure 4A), increased weight loss (Figure 4B), and higher lung viral titers on day 3 post-infection (Figure 4C) than those treated with their wild-type counterparts. Additionally, replacing the human Fc region with murine Fc demonstrated slightly improved protection, as shown by the negligible weight loss (Figure S4A-B). Collectively, our results highlight the importance of Fc-mediated effector functions for the in vivo protection activity of AG2-G02 and 2F02. Consistently, both AG2-G02 and 2F02 could elicit antibody-dependent cell-mediated cytotoxicity (ADCC) activity in vitro (Figure S5).”

Reviewer #2

In this manuscript, Akshita B. Gopal. et al, characterized two group 2 HA stem antibodies AG2-G02 and 2F02. Both antibodies had minimal neutralization activity in vitro but could offer prophylactic protection in vivo. Cryo-EM structures revealed the binding details, and AG2-G02 binds to the lower stem epitope while 2F02 binds to the central stem epitope. Further sequence analysis showed that the natural evolution of HA2 position 32 restricted the binding of AG2-G02 to recent human H3N2 HAs and influences the binding of human plasma samples. Overall, these findings advance our understanding of the antigenicity of HA stem, and the manuscript is well written, and data interpretation is matching the conclusion. However, there are several concerns regarding the presented data before publication.

Response: Thank you for the positive comments.

Questions:

1. Both antibodies have minimal neutralization activity in vitro, but could offer protection in vivo, additional experiments, such as ADCC, are needed to reveal the protection mechanism of these two antibodies.

Response: We have performed additional in vivo experiments as well as an ADCC reporter assay to analyze the protection mechanism of AG2-G02 and 2F02. The results are presented in Figure 4, Figure S4, and Figure S5, and described in the Results section of the revised manuscript:

Lines 176-186: “To investigate the role of Fc-mediated effector functions in the in vivo protection activity of AG2-G02 and 2F02, a LALA-PG variant was generated for each antibody to eliminate their Fc-mediated effector functions⁴⁴. Mice treated with the LALA-PG variants of both AG2-G02 and 2F02 exhibited reduced survival (Figure 4A), increased weight loss (Figure 4B), and higher lung viral titers on day 3 post-infection (Figure 4C) than those treated with their wild-type counterparts. Additionally, replacing the human Fc region with murine Fc demonstrated slightly improved protection, as shown by the negligible weight loss (Figure S4A-B). Collectively, our results highlight the importance of Fc-mediated effector functions for the in vivo protection activity of AG2-G02 and 2F02. Consistently, both AG2-G02 and 2F02 could elicit antibody-dependent cell-mediated cytotoxicity (ADCC) activity in vitro (Figure S5).”

Additionally, we have performed another set of neutralization assays using an alternative protocol as suggested by Reviewer #1 (see our response to comment 1 from Reviewer #1). This protocol reveals stronger neutralization activity for AG2-G02 and 2F02 than what we have reported in our initial submission. As a result, we have also made the following conclusion in the Results section of the revised manuscript:

Lines 186-188: “Nevertheless, the neutralization of AG2-G02 and 2F02 also contribute to their in vivo protection activity, given that their LALA-PG variants were still able to confer protection (Figure 4A-C).”

Similarly, the abstract is modified accordingly:

Lines 34-35: “Both antibodies offer protection in vivo via neutralization activity and Fc-mediated effector functions.”

2. Please give an explanation why 2F02 does not have minimal neutralization activity, compared to previously reported neutralizing antibodies targeting the central stem.

Response: As suggested by reviewer #1, we believe such difference in neutralization activity is due to the difference in how the microneutralization assay was performed. Specifically, while the protocol in the previous version of our manuscript removed the antibodies after one hour of infection, we realize that this practice differs from a published protocol for HA stem antibodies, which keeps the antibodies in the medium for the entire duration of the neutralization assay (PMID: 19079604, PMID: 21737702, PMID: 24335589). Therefore, we have performed another microneutralization assay following the published protocol. Both AG2-G02 and 2F02 showed more potent neutralization activity in this new experiment. These data are added as Figure 3B in the revised manuscript. Please also see our response to comment 1 from Reviewer #1.

3. The authors should further clarify the important implications of their findings for the development of broadly protective influenza vaccines.

Response: In the revised manuscript, the following sentences are included to clarify the important implications of our findings for the development of broadly protective influenza vaccines:

Lines 245-248: “Since central stem antibodies have larger cross-reactivity breadth but weaker neutralization potency compared to lower stem antibodies²⁶, broadly protective influenza vaccines that simultaneously elicit high levels of both types would likely have superior protection efficacy and breadth to those that do not.”

Lines 255-258: “This observation suggests the possibility of minimizing the interference between antibody responses targeting lower stem and central stem epitopes, which should be explored in the future design of broadly protective influenza vaccines.”

4. Female BALB/c mice at 6 weeks old (n = 4-5 per group) were used for the prophylactic protection experiments. In ordinary experiments, 5 or more mice were needed, and the small number of mice might affect the accuracy of the results.

Response: We have increased the group size to 5-7 mice per group in the revised manuscript, thereby strengthening the accuracy of the results. See Methods section:

Line 565: “Female BALB/c mice at 6 weeks old (n = 5-7 per group) ...”

5. In the methods section, “The purified H3N8 full-length, uncleaved protein” should be “The purified H3N8 ectodomain, uncleaved protein”.

Response: We thank the reviewer for pointing this out. This is corrected in the revised manuscript.

6. The purification process of HA ectodomain proteins is not clearly mentioned.

Response: The purification process of the HA ectodomain proteins is now detailed in the Methods section of the revised manuscript:

Lines 393-402: “Soluble His-tagged recombinant ectodomains of H3N2 A/Darwin/6/2021 HA, H4N6 A/mallard/Alberta/455/2015 HA, H3N8 A/mallard/Alberta/362/2017 HA, as well as H1 and H3 stem constructs, were purified from the supernatant using Ni Sepharose Excel resin (Cytiva) for affinity chromatography. The resin was washed with two column volumes each of wash buffer 1 (20 mM sodium phosphate, 500 mM NaCl, 20 mM imidazole, pH 7.4) and wash buffer 2 (20 mM sodium phosphate, 500 mM NaCl, 40 mM imidazole, pH 7.4) to remove non-specific proteins. The proteins were then eluted using a high-imidazole buffer (20 mM sodium phosphate, 500 mM NaCl, 200 mM imidazole, pH 7.4) following a 10-minute incubation. Subsequently, size exclusion chromatography was performed on a HiLoad 16/100 Superdex 200 prep grade column (Cytiva) in 20 mM Tris-HCl (pH 8.0) with 100 mM NaCl.”